# Optimization of Flavonoid Extraction from *Salix babylonica L.* Buds, and the Antioxidant and Antibacterial Activities of the Extract

**DOI:** 10.3390/molecules27175695

**Published:** 2022-09-03

**Authors:** Peng Zhang, Yuwen Song, Hongling Wang, Yujie Fu, Yingying Zhang, Korotkova Irina Pavlovna

**Affiliations:** 1College of Life Engineering, Shenyang Institute of Technology, Fushun 113122, China; 2Primorskaya State Academy of Agriculture, Ussuriisk 692510, Russia; 3College of Animal Science and Veterinary Medicine, Shenyang Agricultural University, Shenyang 110866, China

**Keywords:** Willow Buds, total flavonoids, extraction technology, antibacterial, antioxidation, chemical component

## Abstract

The present study was designed to evaluate the chemical extraction, chemical composition, and antioxidant and antibacterial properties of the total flavonoids in Willow Buds (TFW). We investigated the optimal extraction of TFW using response surface methodology (RSM). Chemical compounds were analyzed using Q-Orbitrap LC–MS/MS. The DPPH radical scavenging capacity, hydroxy radical inhibitory ability, and superoxide anion radical inhibitory ability were explored to determine the antioxidant properties of flavonoid extractions. The antibacterial effect was assessed via minimal inhibitory concentration. The results demonstrated that the optimal extraction conditions were an ethanol concentration of 50%, a time of 35 min, and a liquid/material ratio of 70:1 mL/g. Under these conditions, the yield of TFW was 7.57%. Eight flavonoids, a phenolic glycoside, and an alkaloid were enriched in the Willow Buds. The TFW exhibited significant antioxidant activity, with IC_50_ values of 0.18-0.24 mg/mL and antimicrobial activity against *Escherichia coli*, *Salmonella enterica*, *Staphylococcus aureus*, and *Streptococcus pneumoniae*. TFW may be explored as potential and natural compounds in food and pharmacological applications.

## 1. Introduction

Flavonoids are an important class of secondary plant metabolites and a class of natural beneficial chemicals [1]. Flavonoids have attracted extensive attention due to their anti-inflammatory, antioxidant, antibacterial and antitumor properties [2,3,4]. For instance, flavonoids such as quercetin, kaempferol, and rutin [5,6,7] have various pharmacological activities and are applied in the treatment of various diseases. Therefore, it is necessary to explore optimized methods for the extraction of flavonoids.

Modern extraction technology has the advantages of a high extraction efficiency, purity, and product quality, as well as simple operation and less solvent residue than traditional extraction processes. In recent years, supercritical fluid extraction, ultra-micro-grinding technology, and ultrasonic extraction have been commonly used—especially ultrasonic extraction, which is easy to operate and has a short cycle. Therefore, in this study, we aimed to use ultrasound-assisted extraction to enrich flavonoids and optimize their extraction process conditions using response surface methodology (RSM).

*Salix babylonica L.* is widely distributed all over the world, and its leaves are rich in various compounds such as flavonoids, terpenes and phenols [8]. Studies have shown that *Salix babylonica L.* has a variety of pharmacological activities, including antioxidant, anti-tumor, anti-inflammatory, and antibacterial activities [9,10]. Researchers have extracted the components of *Salix babylonica L.* with hydroalcoholic maceration and found that these pharmacological effects are related to rutin, luteolin glycosides, 3-dioxane-4-(hexadecyloxy)-2-pentadecyl, and kaempferol [8,11,12]. The new buds of *Salix babylonica L.* (Willow Buds) are an edible wild vegetable with high nutritional and medicinal value. In folk medicine, it is commonly used to fight microbial infection, reduce inflammation and improve immunity. However, its extraction conditions, chemical composition, and pharmacological activities are poorly understood. Therefore, in this study, the total flavonoids from *Salix babylonica L.* buds were extracted via an ultrasound-assisted extraction method (UAE), their chemical constituents were identified, and their antioxidant and antibacterial activities were evaluated.

## 2. Materials and Methods

### 2.1. Instruments and Reagents

An UltiMate 3000 RS Ultra Performance Liquid Chromatograph (UPLC) and Q Exactive liquid chromatograph-mass spectrometer (LC-MS) were obtained from Thermo Fisher Scientific (Waltham, MA, USA). A 7230G ultraviolet spectrophotometer was purchased from YOKE Instrument (Shanghai, China). An ultrasonic cleaner (KQ3200DE) was purchased from Kunshan Ultrasonic Instrument (Jiangsu, China). Rutin (purity > 97.0%, MB5118) and vitamin C (VC, purity > 99.0%, MB4168) standard substances were obtained from Meilun (Dalian, China). Doxycycline (K0131507, HPLC purity > 98.0%, CLSI) and levofloxacin (130455-20116, HPLC purity > 98.0%, CLSI) were purchased from the China Institute of Veterinary Drug Control Co., Ltd (Beijing, China) and China National Institutes for Drug Control Co., Ltd (Beijing, China), respectively. DPPH (A153-1-1), a hydroxyl free radical assay kit (A018-1-1), and an inhibition and produce superoxide anion assay kit (A052-1-1) were purchased from Jiancheng Bioengineering Institute Co., Ltd (Nanjing, China).

Standard strains of *Escherichia coli* (ATCC 25922), *Salmonella enterica* (ATCC 51812), *Staphylococcus aureus* (ATCC 25923) and *Streptococcus pneumoniae* (ATCC 49619) obtained from the American Type Culture Collection (Rockville, MD, USA) were used in this study.

### 2.2. Samples and Processing

The Willow Buds of *Salix babylonica L.* (~1 cm) were collected in the Shenyang Institute of Technology from March 2020 to April 2020, and they were authenticated by Professor Junfan Fu of the College of Life Engineering, Shenyang Institute of Technology. After washing, the collected Willow Buds were naturally dried and ground into fine powders for further experiments.

### 2.3. Experimental Design

#### 2.3.1. Single-Factor Experiments

The ambient temperature-dried Willow Buds were extracted via UAE. Three independent factors–ethanol concentration (X1), ultrasonic extraction time (X2), and solvent-to-material ratio (X3)–were used to analyze and compare the extraction efficiency of the total flavonoids in Willow Buds (TFW). The range of all variables were as follows: X1 of 30%, 40%, 50%, 60%, and 70%; X2 of 15, 20, 25, 30, and 35 min; and X3 of 30:1, 40:1, 50:1, 60:1, and 70:1 mL/g.

#### 2.3.2. Optimization of Extraction Conditions by Box–Behnken Design (BBD)

Based on the abovementioned results, the BBD was selected to guide the experimental design with three variables at three levels (Table 1) for 17 randomized experiments (Table 2), and the experimental data were obtained with response surface methodology (RSM).

#### 2.3.3. Determination of TFW Content

The content of TFW was measured with the aluminum nitrate–sodium nitrite–sodium hydroxide colorimetric method, as previously described [13]. Rutin was used as a standard chemical. In brief, rutin (0.01 g) was weighed with an analytical balance and dissolved in 30% ethanol. NaNO_2_ (5%, 0.8 mL) was then added into the rutin solution (10 mL, 0.1 mg/mL). After 6 min, Al(NO_3_)_3_ (10%, 0.8 mL) was added into the mixture. Another 6 min later, NaOH (1 mol/L, 10 mL) was added. The mixture was set to a final volume of 25 mL using 30% ethanol. Following incubation for 15 min, the absorbance was measured at 510 nm. The content of rutin was 0−0.04 mg/mL. The TFW solutions (1.0 mL) extracted under different conditions were diluted 50 times, distilled water was used as a blank control, and the absorbance was measured at 510 nm. The content of TFW was calculated according to the following regression equation: y = 9.6607x − 0.002 (R^2^ = 0.9995), y: absorbance, x: content. The yield of TFW was calculated with the equation given below:(1)$$Extraction\;yield\;\left(mg/g\right)=\frac{C\times V\times N}{m}\times 100\%$$
where *C* represents the concentration of TFW (mg/mL), *V* represents the total volume of the TFW filtrate (mL), *N* represents the dilution ratio, and *m* represents the weight of Willow Buds (g).

### 2.4. Component Analysis and Detection Condition

The extracts (200 mg) obtained with UAE using the abovementioned optimal extraction conditions were dissolved in 1 mL of a methanol: water solution (80:20, *v*/*v*). After being ground with zirconium dioxide for 3 min, samples were centrifuged with 2 × 10^4^× *g* for 10 min at 4 °C. The supernatants were filtered with a 0.22 µm membrane and collected for UPLC and LC-MS analyses.

Our UPLC and LC-MS methods used were similar to those previously described [14,15]. In brief, UPLC analysis was performed using an RP-C18 column (150 × 2.1 mm, 1.8 µm; Welch, Palo Alto, CA, USA). The column was maintained at 35 °C and evaporated at a flow rate of 0.30 mL/min. The mobile phase consisted of water with 0.1% formic acid (A) and acetonitrile with 0.1% formic acid (B). The elution details are shown in Table 3. The needle wash was methanol, the autosampler temperature was 10 °C, the autosampler syringe height was 2.00 mm, and the automatic injection volume was 5.00 µL.

LC-MS analysis was carried out under a positive/negative electrospray ionization source (ESI) switching pattern. The condition parameters were as follows: detection method, full mass/dd-MS2; resolution, 70000 (full mass), 17500 (dd-MS2); scan range, 150.0–2000.0 *m*/*z*; spray voltage, 3.8 kV (positive); data acquisition time, 30 min; collision gas, high-purity argon (≥99.999%); sheath gas, nitrogen (≥99.999%); and auxiliary gas, nitrogen (≥99.999%) held at 350 °C. The capillary temperature was kept at 300 °C.

### 2.5. Antioxidant Capacity Assay

The determination of DPPH, hydroxyl free radical (·OH), and superoxide anion (O_2_–·) was carried out according to the manufacturer’s instructions. We dissolved 1 g of the extract in distilled water and then adjusted the volume to 1.0 mL to obtain a concentration of 1 g/mL. Different final concentrations of samples (62.5, 125, 250, 500, and 1000 µg/mL; 400 µL) were mixed with the DPPH solution (600 µL) and incubated for 30 min at 25 °C in the dark. VC was used as a positive control. The absorbance for *DPPH* radical scavenging activity was recorded at 517 nm. The *DPPH* radical scavenging activity was calculated as follows: (2)$$DPPH\;radical\;scavenging\;activity\;\left(\%\right)=1-\frac{\left(As-Ar\right)}{Ao}\times 100$$
where *As* represents the absorbance of sample after reacting with *DPPH*, *Ar* represents the absorbance of the sample solution after reacting with absolute ethanol, and Ao represents the absorbance of the ultrapure water with DPPH. IC_50_ was defined as the concentration of the TFW extraction, which reached a 50% scavenging effect of the DPPH free radicals.

Similarly, different final concentrations of samples (62.5, 125, 250, 500, and 1000 µg/mL; 200 µL) were incubated for 20 min at room temperature. VC was used as a positive control. The absorbance for ·OH inhibition was measured at 550 nm. The ·OH inhibition ration was calculated using the below equation:(3)$$\cdot \mathrm{OH}\;\mathrm{inhibition}\;\mathrm{ratio}(\%)=\frac{\left(Ar-As\right)}{Ar}\times 100$$
where *As* represents the absorbance of the sample solution after reacting with the ·OH solution and *Ar* represents the absorbance of the sample solution after reacting with absolute ethanol. IC_50_ was defined as the concentration of the TFW extraction, which reached a 50% inhibition effect of ·OH. 

We dissolved 1 g of the extract in distilled water and then adjusted the volume to 1.0 mL to obtain a concentration of 1 g/mL. Different final concentrations of samples (31.25, 62.5, 125, 250, and 500 µg/mL; 0.05 mL) were shaken and stood for 10 min. VC was used as a positive control. The absorbance for O_2_−· inhibition was measured at 550 nm. The O_2_−· inhibition ration was calculated using the below equation:(4)$$o_{2}^{-}\cdot \;\mathrm{inhibition}\;\mathrm{ratio}\left(\%\right)=\frac{\left(Ar-As\right)}{Ar}\times 100$$
where *As* represents the sample solution after reacting with the O_2_−· solution and Ar represents the sample solution after reacting with absolute ethanol. IC_50_ was defined as the concentration of the TFW extraction, which reached a 50% inhibition effect of the O_2_−· free radicals.

### 2.6. Antibacterial Capacity Assay

The antibacterial capacity was measured using the double dilution method following a previous report [16]. In brief, the sample solution (500 µL) in 1% DMSO and the bacterium solution (500 µL) were mixed at 37 °C for 24 h. Then, a doxycycline and levofloxacin solution were used as a positive control, and a 1% DMSO solution was used as a negative control. The experiments were repeated 3 times. The lowest concentration that was not visually cloudy was identified as the minimal inhibitory concentration (MIC).

### 2.7. Statistical Analysis

Data are represented as mean ± standard deviation (SD). The statistics in the RSM experiment were analyzed using Design Expert 8.0.6 software (Trial Version 8.0.6, State, Inc., Minneapolis, USA). The other data were analyzed using the SPSS 17.0 software (SPSS Inc., Chicago, IL, USA). A *p*-value < 0.05 was regarded as statistically significant. Each experiment was carried out in triplicate.

## 3. Results and Discussion

### 3.1. Effects of Ethanol Volume Fraction on the Yield of TFW

It has been reported that the polarity of the solvent is an important factor affecting the extraction rate of flavonoids, and similar polarities lead to higher extraction rates. Ethanol and flavonoids have similar polarities [17], so ethanol was selected as a suitable extraction solvent. Additionally, Krongrawa et al. reported that the optimal extraction conditions of flavonoids from *Kaempferia parviflora* Rhizomes were an ethanol concentration of 54.24%, a time of 25.25 min, and a liquid/material ratio of 49.63 mL/g [18]. According to references and the results of a pre-experiment, a series of parameters were selected in this study. First, different volume fractions of the ethanol solution (30%, 40%, 50%, 60% and 70%) were prepared to evaluate the extraction efficiency of TFW. The solvent-to-material ratio was set as 50 mL/g, and the extraction time was set as 25 min. As shown in Figure 1, the yield of TFW was increased along with the increase in ethanol concentration (X1, %) until the volume fraction of the ethanol solution reached 40%. Furthermore, the TFW yield decreased when the concentration of ethanol was higher than 40%. Considering the rule of like dissolving like, the polarity of the ethanol solution was associated with its volume fraction and flavonoid compounds possessed a better polarity. The results shown in the X1 line in Figure 1 chart demonstrate that an ethanol solution volume fraction of 40% could be an optimal condition for TFW extraction. 

### 3.2. Effects of Ultrasonic Extraction Time on the Yield of TFW

The effect of extraction time on TFW yield is illustrated in the X2 line chart of Figure 1. The other single factors were a 50% ethanol concentration and a 50 mL/g solvent-to-material ratio. The results showed that the yield of TFW reached its highest level when the extracting time peaked at 30 min, and then it gradually declined with increasing time. A main explanation could be that shorter extracting times resulted in the incomplete extraction of TFW and the longer extracting times might have increased energy consumption, worsened flavonoid stability, and accelerated solvent evaporation or decomposition. These results indicate that an extraction time of 30 min might be the best of the studied conditions.

### 3.3. Effects of Solvent-to-Material Ratio on the Yield of TFW

Finally, we tested the optimal ratio of the solvent to material for TFW extraction when the concentration of the ethanol solution was 50% and the extracting time was 25 min. As shown in the X3 line chart of Figure 1, the TFW yield gradually increased along with the increasing solvent-to-material ratio, which reached its peak at 6.5% with a 70:1 mL/g relative solvent-to-material ratio. Considering concentration limitations and experiment costs, a relative solvent-to-material ratio of 70: 1 mL/g could be the most appropriate of studied conditions. 

Thus, following the results of the single-factor TFW experiments, three levels of the three variables—ethanol concentration (30%, 40% and 50%), ultrasonic extraction time (25, 30 and 35 min), and solvent-to-material ratio (50:1, 60:1, and 70:1 mL/g)—were prepared for further RSM analysis. 

### 3.4. Optimization of Extraction Conditions by BBD 

The RSM results are shown in Table 2. The final equation was as follows: Y = 5.84 + 0.58X1−0.21X2 + 1.34X3 + 0.42X1X2 + 0.28X1X3 + 0.10X2X3 − 0.42X12 − 0.095X22 − 0.24X32. The determined coefficient R^2^ = 0.9436 of the final equation indicated that the model fitted well with the experiments, and it could be used to replace the real point of the test to analyze experimental results.

The BBD experimental results for TFW extractives were utilized for variance analysis, as shown in Table 4. Considering that *p*-values were defined to evaluate the statistical difference of the regression coefficients, the determined results (F = 13.02, *p* = 0.0014) of regression models using ANOVA indicated a significant statistical difference when using this model. The X1 and X3 parameters exhibited very significant effects on the response values. In total, the effect of three variables on the extraction rate of TFW was as follows: X3 > X1 > X2.

The parameters (F = 2.10, *p* = 0.2428) found when using the lack-of-fit analysis indicated that the accidental factors, including experimental errors, had no significant effects on experimental results. In addition, the 12.88% of variations could not be explained by this model, as evidenced by RAdj2 = 0.8712. Taken together, this model could be considered to analyze and predict the optimal extraction conditions of TFW. 

The effects of three independent variables—ethanol concentration (X1), ultrasonic extraction time (X2) and the solvent-to-material ratio (X3)–on the yield of TFW extracts were analyzed with Design-Expert 8.0.6 software. The response surface plots are shown in Figure 2a–c. The yield of TFW increased along with increased levels of each of the three independent factors in a certain range. The optimal conditions for TFW extractives were as follows: 50% ethanol concentration, 35 min of ultrasonic extraction time, and a solvent-to-material ratio of 70: 1 mL/g. The maximum yield of TFW at 7.57% was predicted by this model. In addition, the interaction between ethanol concentration (X1) and the solvent-to-material ratio (X3) had the most significant effect on the yield of TFW.

BBD can be used to optimize the extraction of flavonoids by considering the interaction between different factors and shortening the time for screening extraction conditions. Traditional single-factor experiments only involve one variable factor, and the obtained screening data are not reliable enough. On the basis of single-factor experiments, the application of RSM can not only show the influence of interaction terms on the extraction rate of flavonoids but also analyze the accuracy and reliability of data [19,20]. We performed experiments with the optimal parameters, and the yield at 7.55% was identified to be an approximate value in comparison to the predicted results. This result supported the idea that this model could be used to reliably predict experimental values. 

### 3.5. Chemical Component Analysis of TFW Extractives

The chemical components of TFW were acquired with LC–MS and aligned to different databases, including mzCloud, mzVault, and ChemSpider, for characterization. In addition, to screen out the most abundant constituents for more investigations in the future, compounds with a spectral peak area of more than 1 × 10^8^ were analyzed. The Rt, [M–H]−, MS/MS [M–H]−, [M+H]−, MS/MS [M+H]−, calculated mass, and formula of each component are listed in Table 5. The primary and secondary mass spectral profiles of each component are illustrated in Figure 3. The fragment ions of compound **1** and **2** were compared with previously reported data [21,22], and the diagnostic ions at *m*/*z* 138.1 and 463.1 could be assigned to trigonelline and isoquercitrin, respectively. Similarly, the fragments appearing at *m*/*z* 609.1 and 303.0 could be attributed to rutin (compound **3**) [23] and quercetin (compound **4**) [22], respectively. The peaks at *m*/*z* 289.1 and 287.1 matched those of catechin and kaempferol (compound **5** and **6**), respectively [22,23]. Additionally, the fragments at *m*/*z* 447.1 and 317.1 were considered to be characteristic fragments of astragalin and isorhamnetin (compound **7** and **8**), respectively [22,23]. The observed fragments at *m*/*z* 304.1 and 271.1 belong to salicin (compound **9**) [24] and naringenin (compound **10**), respectively [23]. Among the top ten compounds, there was one alkaloid and one phenolic glycoside; the remaining eight were flavonoids. They all have antioxidant and antimicrobial effects, among which rutin [25] and quercetin have the most relative strong activities [26].

### 3.6. Antioxidant Capacity

Researchers have developed several methods to determine in vitro antioxidant capacity (with an increased focus on natural antioxidants), such as by using total oxygen radical scavenging capacity, reducibility methods, ABTS radical scavenging capacity, DPPH radical scavenging capacity, ·OH free radical scavenging capacity, O_2_–· free radical scavenging capacity, and a lipid peroxidation method. The present work was aimed to determine the potential antioxidant activity of TFW through the measurement of DPPH, O_2_–·, and ·OH. The linear fitting equations for the relationships between DPPH, O_2_–· or ·OH with TFW concentration are as follows: Y (DPPH) = –77.383x^2^+133.81x+22.652; Y (O_2_–·) = –203.3x^2^ + 189.86x + 21.264; Y (·OH) = –144.08x^2^ + 227.86x + 1.3191. As shown in Figure 4, TFW significantly increased the DPPH free radical scavenging capacity and O_2_–· and ·OH inhibition capacity in a concentration-dependent manner. The IC_50_ value (mg/mL) was used to represent the antioxidant capacity. According to the equations shown above, the IC_50_ values for DPPH, O_2_–· and ·OH were 0.20, 0.18, and 0.23 mg/mL, respectively, as shown in Table 6. Thus, the results demonstrated that the TFW extracts displayed the radical scavenging capacity of DPPH and the inhibitory properties for O_2_–· and ·OH. The antioxidant activities of the TFW extractives were consistent with those of previous studies regarding flavonoid extractions from *Saussurea involucrate* and *Semen Oroxyli* [27,28]. Importantly, our data showed that TFW exhibited a more significant inhibitory effect on O_2_–· than DPPH radical scavenging and ·OH inhibition, which was similarly to the antioxidant properties of VC. This difference may have been due to the great difference in the DPPH radical scavenging ability and O_2_–· and ·OH inhibition abilities of the different components. Consistent with our results, the antioxidant capacity of flavonoids from tartary buckwheat bran was shown to have a stronger ability for DPPH radical scavenging than ·OH and O_2_–· [29]. Altogether, our work has demonstrated the in vitro antioxidant capacity of TFW to a certain extent. Although previous studies have illustrated the antioxidant effects of willow bark and salix leaf extracts [30,31], our results regarding Willow Buds provide further evidence for the medical value of salix.

### 3.7. Antibacterial Capacity

To evaluate the antibacterial effect of TFW extractives, the *Salmonella enterica*, *Streptococcus pneumoniae*, *Escherichia coli* and *Staphylococcus aureus* strain were used as they are ubiquitous in the natural environment and can cause infections under certain conditions. For instance, a *Streptococcus pneumoniae* infection can cause pneumonia, lower respiratory tract infection, and other diseases [32,33]. *Salmonella* endangers food hygiene and safety [34]. These problems seriously threaten global public health security. Therefore, they were chosen as test microorganisms. Doxycycline and levofloxacin were utilized as the positive control to confirm the accuracy and reliability of experimental measurements. As shown in Table 7, the TFW extracts showed antibacterial effects against the four strains, in which the MIC of *Staphylococcus aureus* was 2.5 mg/mL and that of *Salmonella enterica* was 10 mg/mL. Recently, accumulating traditional Chinese medicines were found to possess antibacterial pharmacological activities, and these were gradually used to replace the usage of antibiotics due to the bacterial resistance. González-Alamilla et al. demonstrated that the chemical compounds of *Salix Babylonica L.* exhibited antibacterial capacities against *Escherichia coli*, *Staphylococcus aureus* and *Listeria monocytogenes*. The best antibacterial property was obtained with luteolin against *Staphylococcus aureus* [11]. The salicin components from the stem bark of *Salix tetrasperma ROXB.* were also found to show evident antibacterial capacity [35]. In 2020, González-Alamilla et al. evaluated a methanolic extract of *S. babylonica* against *Escherichia coli*, *Salmonella typhi*, *Salmonella choleraesuis*, and *P. aeruginosa* and determined an MIC of 100 mg/mL, 25 mg/mL for *S. aureus* and *L. monocytogenes,* and 12.5 mg/mL for *Bacillus subtilis* [36]. Our results suggested that the abundant TFW have antimicrobial effects against Gram-positive and Gram-negative bacteria. They accordingly could be an ideal agent in food processing, beauty care and the clinical antibiotic replacement. 

## 4. Conclusions

In summary, the authors of the present study extracted flavonoids from Willow Buds using an ultrasonic-assisted extraction method for the first time. The optimized conditions for flavonoid extraction were a 50% ethanol concentration, 35 min of ultrasonic extraction time, and a 70:1 mL/g solvent-to-material ratio. The TFW extracts were determined to exhibit in vitro antioxidant and antibacterial activities. We also described the active components (eight flavonoids, a phenolic glycoside, and an alkaloid) present in Willow Buds.

## Figures and Tables

**Figure 1 molecules-27-05695-f001:**
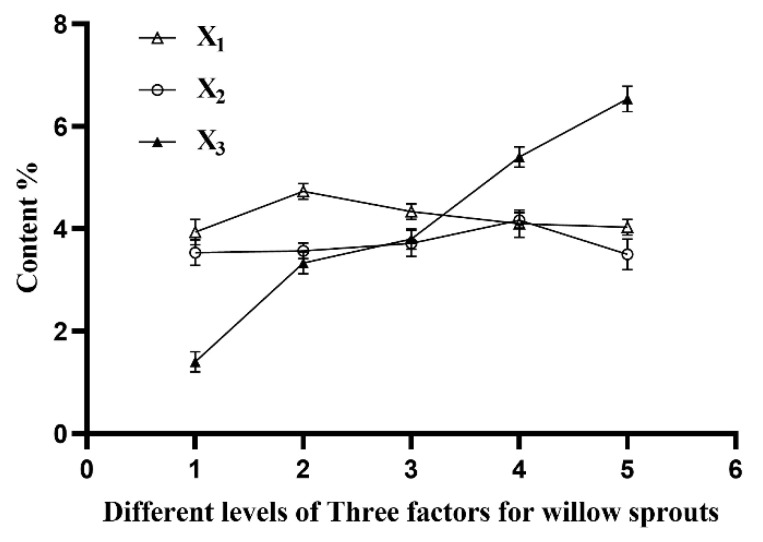
Effects of ethanol concentration (X_1_, %), extraction time (X_2_, min), and ratio of solvent to material (X_3_, mL/g) on the extraction efficiency of TFW.

**Figure 2 molecules-27-05695-f002:**
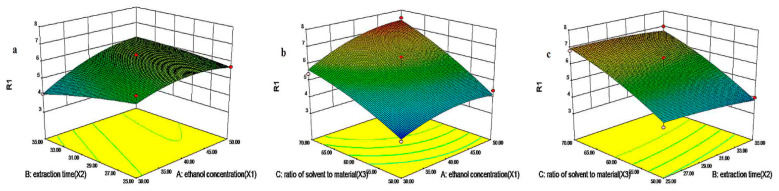
(**a**–**c**) Response surface plots of ethanol concentration (X1, %), extraction time (X2, min), and ratio of solvent to material (X3, mL/g) on the extraction yield of TFW.

**Figure 3 molecules-27-05695-f003:**
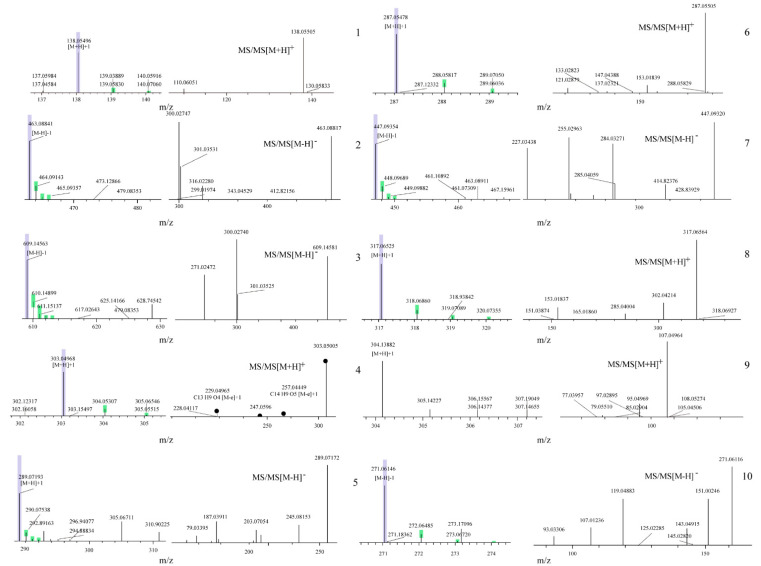
MS and MS/MS spectra profiles of 10 constituents in TFW measured by LC-MS.

**Figure 4 molecules-27-05695-f004:**
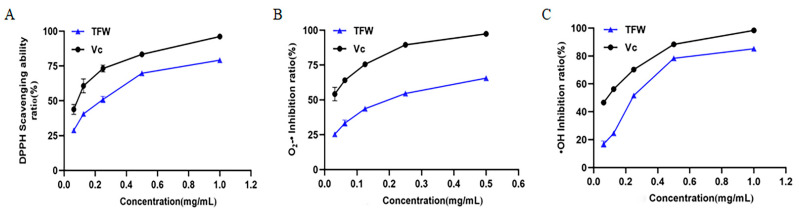
(**A**–**C**) Effect of TFW on DPPH, O_2_–· and ·OH free radicals.

**Table 1 molecules-27-05695-t001:** Independent variables and their levels in BBD for TFW.

Levels	Ethanol Concentration (X1)/(%)	Extraction Time (X2)/(min)	Ratio of Solvent to Material (X3)/(mL/g)
–1	30	25	50
0	40	30	60
1	50	35	70

**Table 2 molecules-27-05695-t002:** Box–Behnken experimental design and the results for the extraction yield of TFW (*n* = 3).

No.	Independent Variable Levels			Response
	Ethanol Concentration (X1)/(%)	Extraction Time (X2)/(min)	Ratio of Solvent to Material (X3)/(mL/g)	
1	30	25	60	5.8
2	50	35	60	5.7
3	40	30	60	5.8
4	50	25	60	5.7
5	40	25	70	6.8
6	40	35	70	7.0
7	30	30	50	3.4
8	30	35	60	4.1
9	50	30	50	4.4
10	40	30	60	6.4
11	40	25	50	4.2
12	30	30	70	5.4
13	40	30	60	5.5
14	50	30	70	7.5
15	40	30	60	5.7
16	40	35	50	4
17	40	30	60	5.8

**Table 3 molecules-27-05695-t003:** Chromatographic gradient elution.

Time (min)	A (%)	B (%)
0	98	2
1	98	2
5	80	20
10	50	50
15	20	80
20	5	95
25	5	95
26	98	2
30	98	2

**Table 4 molecules-27-05695-t004:** Analysis of variance for the BBD experimental results for TFW.

Variables	Sum of Squares	DF	Mean Square	F Value	*p*-Value
Mode	19.5	9	2.17	13.02	0.0014
X1	2.65	1	2.65	15.90	0.0053
X2	0.36	1	0.36	2.17	0.1841
X3	14.31	1	14.31	86.03	<0.0001
X1*X2	0.72	1	0.72	4.34	0.0756
X1*X3	0.30	1	0.30	1.82	0.2195
X2*X3	0.040	1	0.040	0.24	0.6389
X12	0.74	1	0.74	4.46	0.0725
X22	0.038	1	0.038	0.23	0.6473
X32	0.25	1	0.25	1.52	0.2575
Residual	1.16	7	0.17		
Lack of Fit	0.71	3	0.24	2.10	0.2428
Pure Error	0.45	4	0.11		
Cor total	20.66	16			
R2	0.9436				
RAdj2	0.8712				
RPred2	0.4142				
Adeq Precision	12.227				
C.V.%	7.44				

**Table 5 molecules-27-05695-t005:** Identification of the chemical components of TFW extractives.

No.	Rt (min)	[M–H]–	MS/MS [M–H]–	[M+H]–	MS/MS [M+H]–	Calculated Mass	Formula	Proposed Molecule	Reference
1c	2.61	_	_	138.1	110.1, 139.1	137.0	C_7_H_7_NO_2_	trigonelline	[21]
2a	5.22	463.1	299.0, 301.0, 300.0	_	_	464.1	C_21_H_20_O_12_	isoquercitrin	[22]
3a	5.66	609.1	300.0, 301.0, 271.0	_	_	610.2	C_27_H_30_O_16_	rutin	[23]
4a	5.67	_	_	303.0	257.0, 229.1, 247.1	302.0	C_15_H_10_O_7_	quercetin	[22]
5a	5.79	289.1	203.1, 245.1_	_	_	290.1	C_15_H_14_O_6_	catechin	[22]
6a	5.91	_	_	287.1	121.0, 153.0	286.0	C_15_H_10_O_6_	kaempferol	[23]
7a	6.35	447.1	284.0, 255.0, 227.0	_	_	448.1	C_21_H_20_O_11_	astragalin	[22]
8a	6.73	_	_	317.1	302.0, 153.0, 285.0	316.1	C_16_H_12_O_7_	isorhamnetin	[23]
9b	7.25	_	_	304.1	107.1	286.1	C_13_H_18_O_7_	salicin	[24]
10a	7.58	271.1	93.0, 107.0, 119.0, 151.0	_	_	272.1	C_15_H_12_O_5_	naringenin	[23]

a, flavonoid; b, phenolic glycoside; c, alkaloid.

**Table 6 molecules-27-05695-t006:** The results of the antioxidant activities of TFW extractives.

Indicators	Antioxidants	R^2^ of Linear Fit	IC_50_ (mg/mL)
DPPH	TFW	0.9941	0.20
	VC	0.9921	0.08
O_2_–·	TFW	0.9911	0.18
	VC	0.9931	0.03
·OH	TFW	0.9951	0.24
	VC	0.9982	0.08

**Table 7 molecules-27-05695-t007:** The results of the antibacterial activities of TFW extractives.

Bacterial Strain	Doxycycline (µg/mL)	Levofloxacin (µg/mL)	TFW (mg/mL)
*Escherichia coli* (ATCC25922)	0.5	0.0625	5
*Streptococcus pneumoniae* (ATCC49619)	0.0625	2	5
*Staphylococcus aureus* (ATCC29213)	0.25	0.25	2.5
*Salmonella enterica* (ATCC 51812)	8	4	10

## Data Availability

Not applicable.

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
