# Peer review of "Optimization of Flavonoid Extraction from Salix babylonica L. Buds, and the Antioxidant and Antibacterial Activities of the Extract"

_molecules, 2022, doi:10.3390/molecules27175695_

Round 1

Reviewer 1 Report

1. Introduction

Salix babylonica L. buds exhibits various bioactive components and biological activities.

Why were the researchers interested in some flavonoids and alkaloids, as well as antioxidant activity?

2.1. Instruments and reagents

Add the information of the standard substances used.

2.4. Component analysis

Why was methanol:water solution (8:2, v/v) used as an extraction solvent for component analysis?

2.5. Detection condition

Add the references of UPLC and LCMS methods used and data involving system suitability.

If self-developed analytical methods. Please show the method development and validation.

2.6. Antioxidant capacity assay

What kinds of solvents used for antioxidant testing?

Specify the reasons for choosing the test method.

3.1. Optimization of single factor experiments

Explain the reasons for choosing ethanol for extraction.

Explain the reasons for setting the range of ethanol concentration, duration of time scale, and the range of solvent to material ratio.

Edit printing subscription on page 7.

On page no.9

Explain reasons for interest in analyzing 10 substances in Table 5.

Which substance is the most potent in antioxidant and antimicrobial activities (add the refernces)?

3.8. Antibacterial capacity

Add the reasons for choosing Salmonella enterica (ATCC 51812) and Streptococcus pneumoniae as test microorganisms.

Optimization of extraction correlated with antioxidant activities but didn’t show any effects of UAE on antimicrobial activities related to the title “Effect of ultrasound-assisted extraction on the phenolic composition, and antioxidant and antibacterial activities of Salix babylonica L. buds”. It shows only antimicrobial activity of TFW in Table 7.

Reviewer 2 Report

The manuscript is interesting but it is not acceptable in the current state. 

It needs substantial improvement in English language.

The introduction is poor, with general information, while specific information about the constituents of the plant under examination and the extraction procedures reported in literature are missing. Thus, the up to date knowledge about the examined topic is not presented and the need for the present research is not obvious.

The experimental procedure is not well described.

It should be stated if an extract sample was used for TFW analysis, and what is the “total flavonoid powder” that was used for component analysis.

Line 106-107: What do you mean by “ Once assessing … for analysis”?

Line 110-111: The BBD design has a specific combination of the experimental variables. What do you mean by “when determining … at the middle level?

Line 153, 163, 172: please clarify the “sample solution” and concentration in what and how was it obtained.

Results and discussion

Lines 193-198 and 234-237 should be moved to the experimental section.

The authors are encouraged to discuss the differences in optimum conditions obtained by single factor experiments and BBD.

Section 3.6: Comparison with literature data is missing.

Lines 295-297: the equations are not linear. The authors are encouraged to discuss the usefulness of these equations.

Lines 305-312: It would be interesting to correlate the differences in DPPH* OH- and -O2 with the different consistency of the mentioned extracts.

Conclusion

Line 347-348: The statement is not supported by the data of this work.

Some additional suggestions are included in the attached file.

Round 2

Reviewer 1 Report

2.4. Component analysis and 2.5. Detection condition should be merged into one section.

2.6. Antioxidant capacity assay “The extracts (1 g) were dissolved in 1 mL of distilled water, then diluted into different concentrations (62.5, 125, 250, 500, 1000 μg/mL).” …Dissolving 1 g of the extract in 1 mL of distilled water will not yield a concentration of 1 g/mL, whereas dissolving 1 g of the extract in distilled water and then adjusted the volume to 1.0 mL will obtain a concentration of 1 g/mL. Likewise, the lines 151 through 152.

In lines 262-273 and 324, substance names should use lowercase letters.

In lines 312-313, 325-327, and some references, scientific names should be italicized.

In Table 7, ATCC xxxx should be in straight letters.
